# Atomistic Study on the Sintering Process and the Strengthening Mechanism of Al-Graphene System

**DOI:** 10.3390/ma15072644

**Published:** 2022-04-04

**Authors:** Yongchao Zhu, Na Li, Wei Li, Liwei Niu, Zhenghui Li

**Affiliations:** 1Department of Railway Engineering, Zhengzhou Railway Vocational and Technical College, Zhengzhou 451460, China; liwei@zzrvtc.edu.cn (W.L.); 11140@zzrvtc.edu.cn (L.N.); lizhenghui@zzrvtc.edu.cn (Z.L.); 2School of Mechanics and Engineering Science, Zhengzhou University, Zhengzhou 450001, China; zzulina@zzu.edu.cn

**Keywords:** MD, sintering, graphene, composite, strengthening mechanism

## Abstract

The powder metallurgy process of the Al–graphene system is conducted by molecular dynamics (MD) simulations to investigate the role of graphene. During the sintering process, graphene is considered to reduce the pore size and metal grain size based on the volume change and atomic configuration of the Al parts in the composite. Compared with the pure Al system, the space occupied by the same number of Al atoms in the sintered composite is 15–20 nm^3^ smaller, and the sintered composite has about 5000 fewer arranged atoms. Because these models are carefully designed to avoid a serious deformation of graphene in the tension of sandwich-like composite models, the strengthening mechanism close to the experimental theory where graphene just serves to transfer a load can be studied dynamically. The boundary comprising of two phases is confirmed to hinder the motion of dislocations, while the crack grows along the interface beside graphene, forming a fracture surface of orderly arranged Al atoms. The results indicate that single-layer graphene (SLG) gives rise to an increase of 1.2 or 0.4 GPa in tensile strength when stretched in in-plane or normal direction, while bilayer graphene (BLG) brings a clear rise of 1.2–1.3 GPa in both directions. In both in-plane and normal stretching directions, the mechanical properties of the composite can be improved clearly by graphene giving rise to a strong boundary, new crack path, and more dense structure.

## 1. Introduction

Owing to the superior mechanical properties, graphene is naturally expected to work as an effective reinforcement phase to improve metal matrix composite materials. Existing studies have provided quite a number of methods to fabricate metal–graphene composites such as powder metallurgy sintering, composite electroplating, differential speed rolling, etc. In those techniques, powder metallurgy sintering is supposed to be an efficient way to prepare metal–graphene composite in bulk because of the simple process and precise control, drawing considerable attention of researchers recently [1,2,3].

Generally, most of the investigations on sintering metal–graphene composites are performed by experiments. Some focus on problems of graphene dispersion hindering the preparation of a homogeneous structure. Mu et al. applied an electroless plating method to prepare Ni-decorated graphene nanoflakes as a reinforcement to be uniformly dispersed in a Ti matrix, and then spark plasma sintering (SPS) and hot-rolling are followed to gain a titanium matrix composite with an enormous strength increase [4]. Ju et al. prepared graphene-reinforced aluminum (Al) matrix composites with enhanced dispersion and strength, based on an aqueous suspension mixing procedure and SPS technology [5]. Li et al. achieved a densified graphene–Al composite with uniform distribution of reduced graphene oxide via electrostatic interaction and hot pressing [6]. Kumar et al. carried out the dispersion of graphene by ultra-sonication followed by ball milling and adding Al6061 alloy powder, before sintering under inert gas atmosphere. The others pay attention to the rise in mechanical properties with graphene content [7]. Liu et al. revealed that there are great increases of 32% and 43%, respectively, in hardness for Al matrix composites with 0.3 wt% reduced graphene oxide and 0.15 wt% graphene nanosheets [8]. Tian et al. found that the hardness, compressive strength, and yield strength of the composites are improved with the addition of 1 wt% graphene, but further addition of graphene will result in the deterioration in mechanical properties [9]. Shin et al. investigated the strengthening behavior of composite containing 0.7 vol.% few-layer graphene, exhibiting 440 MPa of tensile strength [10]. Rashad et al. reported an aluminum–graphene composite with a rise of 14.7% in yield strength and an increase of 11.1% in ultimate tensile strength, using a semi-powder method [11].

The experimental studies listed in Table 1 indicate that the mechanical properties of the composite will be truly improved by the addition of graphene, and microscopic images revealed that fine grain strengthening and dislocation strengthening can be enhanced by graphene. However, it is difficult to present the structural evolutions in both sintering process and loading progress dynamically, especially in atomic scale. However, the dynamical atomic configurations are readily available in molecular dynamics (MD) simulations, which can trace the motion of each atom. Therefore, many efforts are made to model the sintering process with different metal nanoparticles. Jiang et al. explored the structural evolution and underlying sintering mechanism of aluminum nanoparticles in terms of average displacement, mean squared distance, radius ratio, etc. [12]. Liu et al. revealed the rule of atomic migration and the growth mechanism of the sintering neck with Fe nanoparticles [13]. This means the structural evolution during sintering can indeed be monitored through a computational study. Tavakol et al. investigated the mechanical properties of nanocomposites produced by shock wave sintering of aluminum and silicon carbide nanopowders [14]. Wejrzanowski et al. checked the melting behavior of Al–Si nanolayers with two different thicknesses to explain a noticeable decrease in the melting temperature [15]. He et al. studied the sintering process of graphene nanoplatelet-reinforced aluminum matrix composite powder and the mechanical properties of sintered composites [16]. Kumar et al. explored the orientation of aluminum atoms along with the mechanical properties of an aluminum/graphene nanocomposite, after a cooling process from 2500 K to 10 K [17]. It is illustrated that the MD method is also suitable for investigating the sintering and mechanical behaviors of composite systems.

Nevertheless, most of the MD papers only consider the system of pure metal particles [12,13] or metal nanoparticles [14,15]. The studies discussing the Al–graphene composite system often suffer the drawbacks of neglected details in initial models, where the size of graphene is equal to the plane size of the composite system. Such a sandwich-like structure under tension will give rise to a serious deformation in graphene bearing most of the direct stress, and the tensile strength can be as high as 14 GPa [16]. Although the graphene size in [17] is more reasonable, a crystallization process rather than sintering process is employed to prepare the Al–graphene composite and multilayer graphene has not been discussed. Furthermore, Kvashnin et al. proved that an increase in the width of graphene can actually lead to a substantial increase in the mechanical characteristics by providing efficient load transfer [18]. Therefore, it is significant to examine the metal–graphene system by means of a rational model. Thus, carefully designed models in this paper are created to observe the changes in composite structure, revealing the effects of graphene on the sintering and tensile processes.

## 2. Model Methods

In the sintering systems of Al–graphene composites, MD models shown in Figure 1a contain eight spherical Al nanoparticles with a diameter of 56 Å, and the distance between Al particles is set to 8 Å. A square nanoplatelet of monolayer or bilayer graphene with each side of 64 Å is inserted into the gap between Al particles, as shown in Figure 1a. It is worth mentioning that the crystal orientations of Al particles are different from each other. Periodic boundary conditions (PBC) are applied in each dimension of the simulation box, so the NPT ensemble can be adopted to relax the system at 300 K and 1 atm. After a relaxation time of 100 ps, Al particles will be integrated together with the graphene nanoplatelet embedded naturally, obtaining an initial structure for sintering. 

During the sintering process, the composite structure will be heated from 300 K to 773 K in 300 ps, with the external pressure increasing from 1 atm to 500 atm (50 MPa). Then, the system maintains such temperature and pressure for 300 ps. At last, the external conditions of sintered composite will return to 300 K and 1 atm after a cooling stage of 300 ps. The NPT ensemble is also applied in the entire sintering process. It should be noted that parameter settings above such as the heating and cooling rate are selected according to the similar sintering process of the Al–graphene system in reference [16].

Then, tensile simulation is conducted on sintered composite to examine the mechanical behaviors. Another relaxation process will be carried out at 300 K for 100 ps to eliminate thermal stress before tensile testing. The NVE ensemble is applied to operate quasi-static tension by changing the shape of the simulation box in both in-plane and normal directions, and the temperature is reset by explicitly rescaling their velocities. 

All the MD simulations are carried out by LAMMPS and the generated data are visualized by VMD software. The velocity *Verlet* algorithm is used to solve Newton’s equations of motions and the time step is set to 1.0 femtosecond. The *eam* potential is applied to the interaction between Al atoms [19], the file of which can be obtained from the LAMMPS library, and a simple sintering model of pure Al bulk is carried out to verify this potential file by checking the melting point (Figure 1b). The *airebo* potential is used to link carbon atoms in graphene [20], as commonly applied in [16,21,22] to mimic graphene; the Al–C interaction takes the style of the *morse* potential [23], which is proved to be more suitable for the interface between metal and graphene [24]. 

## 3. Results and Discussion

### 3.1. The Sintering Processes

As is known to all, the powder metallurgy process can promote densification of metal particles by virtue of plastic deformation and material transfer during the sintering, leading to a decline in porosity, a shrinkage in volume, and an increase in density. To validate the MD model in this paper, a sintering process of pure Al particles is operated, firstly, after relaxation on the initial model, followed by monitoring of the changes in system volume and surface morphology. As expected, the system temperature and pressure are successfully controlled during the sintering process (Figure 2a). The black line in Figure 2b indicates that the system volume falls rapidly from 918.2 nm^3^ to 816.9 nm^3^ in the heating process, while the configures in Figure 2c reveal that surface energy may lead to formation of the sintering neck before sintering, and then the sintering neck will grow quickly with the temperature rising, leading to the diminishment of pore size. In the stage of constant temperature and pressure, the bulk volume largely holds steady after a slight decrease from 816.9 nm^3^ to 801.0 nm^3^; the surface morphology of sintered bulk has been basically established and the final drop of volume can be simply attributed to cold shrink in the cooling stage. This change is mainly in line with expectations, which have been reported in many experimental studies [7,9]. 

Based on the pure Al model, Al–graphene composite bulks are sintered to investigate the role of graphene on the powder metallurgy process. The initial structures for sintering are prepared by inserting the graphene nanoplatelet into the gap between Al particles. Firstly, a single-layer graphene (SLG) is placed in the center gap of the pure Al model. It means both sides of the SLG are adjacent to four Al particles, forming a pore containing graphene, as marked in Figure 2d. After sufficient relaxation on this composite model, the same sintering process before will be conducted again to achieve the Al–SLG composite. From the volume change shown by a red line in Figure 2b, it is fairly easy to find that the development of the sintering process seems to be sped up by the addition of graphene; the system volume goes down faster from 995.5 nm^3^ to 806.3 nm^3^ during the heating stage, which keeps a lower than pure Al curve after about 150 ps, and then falls to 777. 4 nm^3^ at constant temperature and pressure. The curves also indicate a possibility of further decline in volume, as reported in [25] with impulse pressure assisted. However, it is enough to study the effect of graphene on the sintering process based on the models here. The final volume is 753.4 nm^3^, which is a little less than 759.6 nm^3^ of the pure Al model, even though there is an extra SLG. Therefore, the same number of Al atoms in the Al-SLG composites occupies a smaller space, as presented in Table 2. These indicate that graphene is conducive to promoting the sintering progress and improving the density of sintered structures, which can be ascribed to the higher binding energy between Al atoms and graphene [16], providing more energy for Al motions approaching graphene. 

To gain insights into the inner structure of sintered bulk, centro-symmetry parameter (CSP) analysis is performed, and the local lattice disorder is tinted in CSP values. Blue balls stand for Al atoms with bulk lattice, red balls represent surface atoms, and other colorful balls are classified as defects such as atoms in dislocation or grain boundary. Therefore, the size of Al grains can be directly determined by color configurations of atomic structures. Since red atoms imply surface atoms appearing inside, the pores can be easily found in the sintered structure. As marked with yellow circles in Figure 3, the pores in the Al–SLG composite are visibly smaller than those in the pure Al model, suggesting that graphene can reduce the size of inner pores, accounting for the change of the system volume above. Because the crystal orientations of Al nanoparticles in initial models are different from each other, polycrystalline structures are expected to be formed naturally in sintered structures, and Al grain sizes can be estimated by the blue areas divided by the green belts in Figure 3. The grain size of the sintered composite seems to be a little smaller than that of the pure Al model, on the basis of inner structures in different views. Especially in the sectional view across graphene, it is easy to find that embedded graphene is obviously bent as the volume shrinks and heavily disturbs the arrangement of near Al atoms, forming dislocations and enlarging the grain boundary. For quantitative analysis on the sintered structures, common neighbor analysis (CNA) is operated to count the number of atoms in local f.c.c. (face-centered cubic) order and h.c.p. (hexagonal close-packed) order. The former is considered as perfect grains, the latter is identified as stacking faults, and both can be regarded as arranged atoms. Distinctly in Table 2, the Al–SLG composite has less atoms of perfect lattice and arranged structures. Therefore, it is safe to say that graphene can promote grain refinement. 

In order to study the effect of multilayer graphene on the powder metallurgy process, SLG in the models is substituted by bilayer graphene (BLG) to gain the Al–BLG composite. Then, the sintering simulation and structure analysis are performed as before. In comparison with the composite structure of SLG, BLG is likely to make little difference in the volume and structure of the Al–graphene composite. Based on Figure 2a and Table 2, the curves of the composite volume are similar, the total number of arranged atoms is almost the same, except for that more Al atoms are classified into the h.c.p. lattice. This is because Al atoms arranged on the honeycomb lattice of graphene happen to form the close-packed plane [26], which determines the local crystal orientation, and the interlayer space in the BLG may lead to a slight mismatch of grains on both sides of the BLG, facilitating the occurrence of more stacking faults or grain boundaries that can be noticed in Figure 3. 

### 3.2. The Tensile Processes

The strengthening mechanism of graphene has been illumined experimentally by some static micro images of composite structures, where high density of dislocations accumulates around the boundary comprising two phases and this boundary is expected to hinder the motion of dislocations effectively [27,28]. It suggests that graphene can improve the adhesion strength of the interface to prevent the transcrystalline break. Here, such dynamical progress can be easily detailed by MD simulations.

Uniaxial tensile simulations are performed here to investigate the enhancement mechanism of graphene in metal matrix composites, and the stretching direction is along the *y* or *z* axis where the tension is roughly parallel or normal to graphene. Then, the stress–strain curve is drawn to compare strength and deformation, and CSP analysis of Al atoms is used to trace the change of inner structure including dislocation motion, propagation of the cavities, and morphology of fracture surface. Figure 4 presents the changes of inner structure under different strains in the tensile process. After relaxation for eliminating thermal stress, there are no visible changes in the pore of the pure Al model, and then the pore grows larger with the motion of dislocations deriving from the increasing strain. However, the original pore in the Al–SLG model seems to be filled with the release of thermal stress. The dislocations (indicated by a red arrow) pile up around graphene gradually under tensile deformation, giving rise to a crack formed near the graphene (marked by a yellow circle), and then the crack propagates along the interface beside the graphene quickly. The entire processes of fractures are detailed by the evolution of atomic structures in the whole range of strain (Appendix A). Due to the strong binding force between Al and graphene, Al atoms on the fracture surface are arranged orderly. This cracking behavior can also be observed in composite models with BLG, just as the crack initiation and fracture appearance are shown in Figure 5.

It is remarkable that quite a few studies simply attribute the wonderful mechanical performance to graphene sharing most of the stress, where the sizes of graphene are the same as the plane size in models. However, the distribution of atom stress in Figure 6 reveals that only a part of the graphene bears tensile loads, and the deformation of graphene lattice structure looks very small, despite that the stress values of the carbon atoms are visibly a little higher than those of the adjacent Al atoms. It means that embedded graphene just plays a functional role in transferring tensile load [29], especially in the model with normal tension where graphene is prone to in-plane bending. Therefore, it is safe to say that graphene may give rise to a distinct improvement in mechanical performance, but it is impossible for the composite to approach the level of graphene itself, unlike other simulations on metal–graphene composites where graphene directly bears a large deformation [16,30,31,32].

The stress–strain curves of models stretched in in-plane and normal directions are drawn in Figure 7. It should be noted that the inverse Hall–Petch relationship will lead to a drop in strength at this scale where metal grains have a size of less than 10 nm. Distinctly, composites with both SLG and BLG exhibit a rise of about 1.2 GPa in tensile strength when stretched in in-plane direction, compared with the pure Al model. However, the Al–SLG composite only shows a slight advantage of 0.4 GPa when stretched in the normal direction. The Al–BLG composite can still achieve an improvement of 1.3 GPa at a larger strain because the crack has not developed along the gap between BLG layers, as depicted in Figure 5d. The interlayer space in the BLG may help to reduce the normal stress concentration to some extent, rather than serve as the crack propagation path. Therefore, we can conclude that graphene can definitively improve the mechanical property of the metal–graphene composite by the enhanced interface described above, regardless of the stretching direction. In addition, the pores in the pure Al model are visibly bigger than that in the sintered composite. Therefore, the enhancement mechanism of the Al–graphene composite can be ascribed to a more compact structure, a two-phase boundary, and a new crack path, all of which are deemed to be contributed by the graphene.

## 4. Conclusions

MD simulations on the powder metallurgy process are performed to study the sintering process and the strengthening mechanism of the Al–graphene system. Details on volume and local crystal structure of Al atoms in a composite, which cannot be measured through the experimental method, indicate that graphene is helpful for promoting the sintering progress, improving the density of sintered structures, and refining metal grain size. The space occupied by the same number of Al atoms in the sintered composite is 15–20 nm^3^ smaller than the pure Al structure, and the amount of arranged Al atoms in the sintered composite is about 5000 fewer too. The effect of graphene in the mechanical property is studied minutely through the tensile process on the sintered composite. The composite with the SLG can exhibit an increase of 1.2 or 0.4 GPa in tensile strength when stretched in in-plane or normal direction, while the composite with the BLG can achieve a clear rise of 1.2–1.3 GPa in both directions. This suggests that graphene is of great importance for composites to improve the mechanical performance whether in in-plane or normal tensile direction. This improvement can be attributed to the effect of graphene, contributing to the formation of two-phase boundary, new crack path, and more compact structure, which can be easily found in view of the atomic configures. 

## Figures and Tables

**Figure 1 materials-15-02644-f001:**
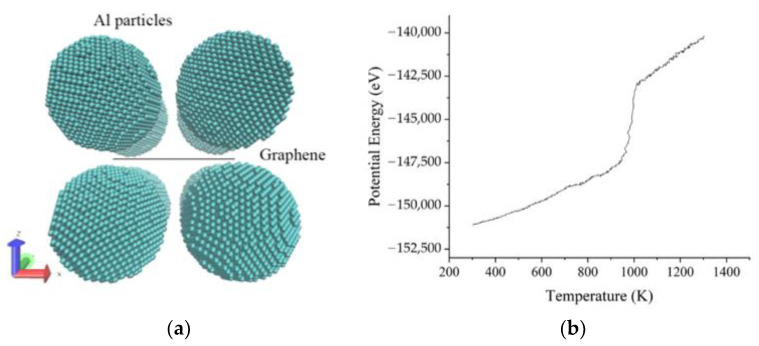
Details of the sintering models. (**a**) Atomistic models for building; (**b**) potential energy of pure Al bulk with respect to temperature.

**Figure 2 materials-15-02644-f002:**
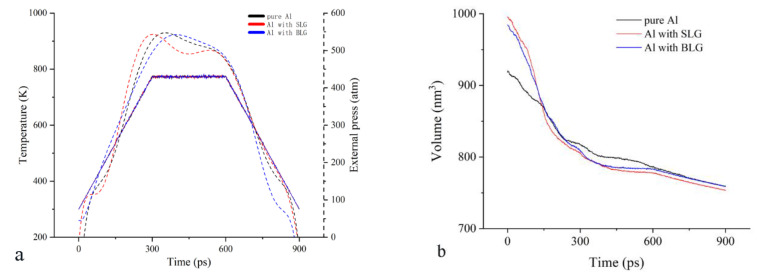
The changes in the sintering process. (**a**) The system volume; (**b**) the system temperature and pressure; and atomic configurations of the pure Al model (**c**) and Al-SLG model (**d**) at different sintering times.

**Figure 3 materials-15-02644-f003:**
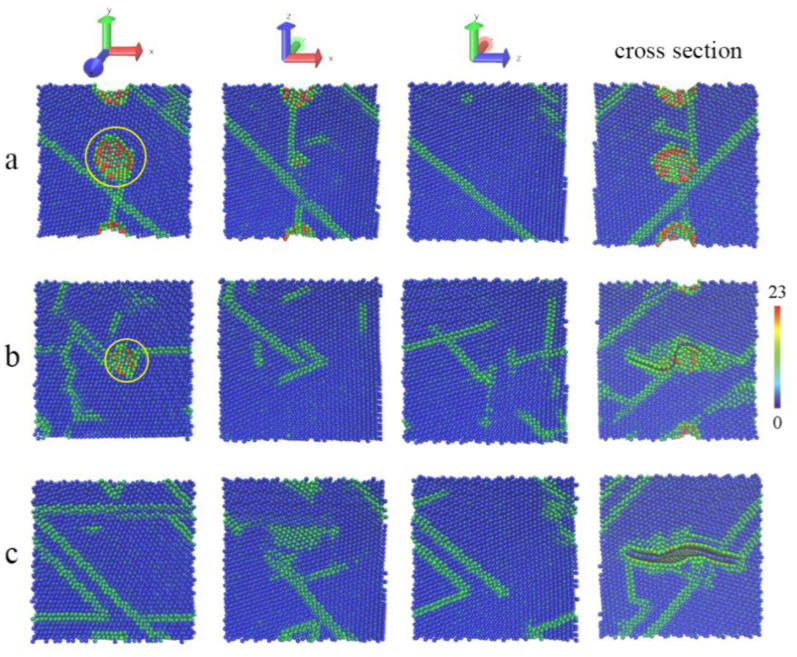
The atomic configures of sintered models in different views. (**a**) The pure Al structure, (**b**) the structure of the composite with SLG, (**c**) and the structure of the composite with BLG.

**Figure 4 materials-15-02644-f004:**
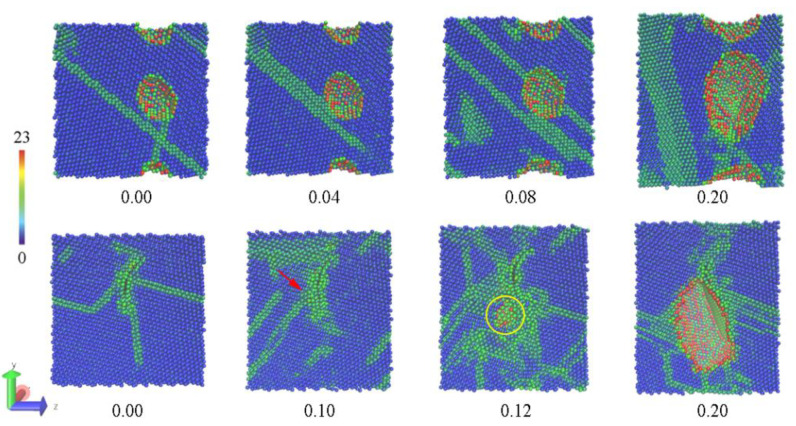
The changes in the inner structure under different strains in the tensile process. The pure Al structure is shown in the upper row, while the structure of the composite with SLG is shown in the lower row.

**Figure 5 materials-15-02644-f005:**
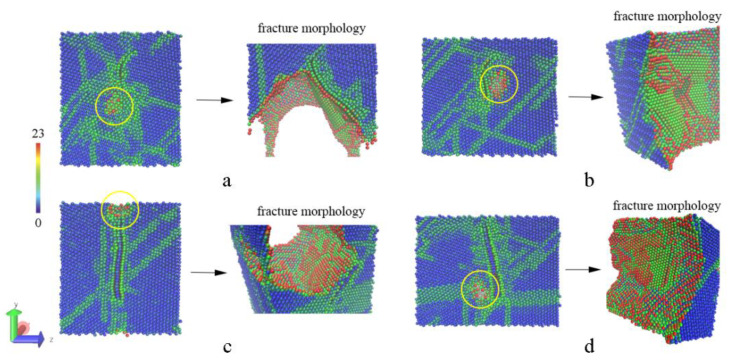
The atomic details of initial cracks and fracture surfaces. (**a**,**b**) The composite with SLG is stretched in Y and Z directions; (**c**,**d**) the composite with BLG is stretched in Y and Z directions.

**Figure 6 materials-15-02644-f006:**
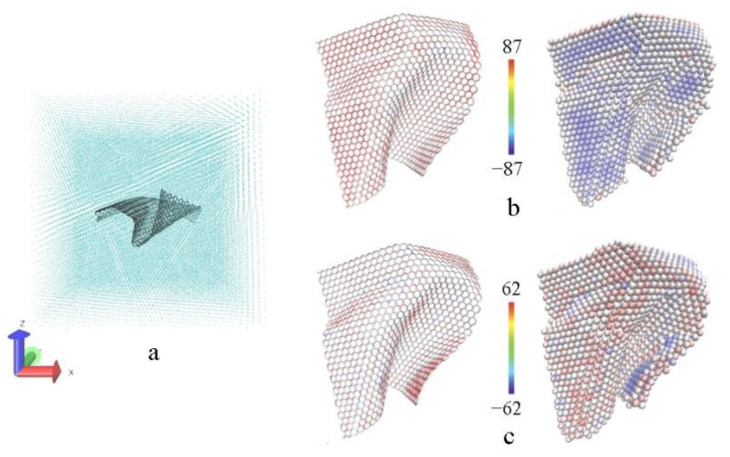
The atomic details of SLG in the tensile process. (**a**) SLG in sintered structure before stretching; the stress distribution in SLG under maximum stress stretched in the *y* axis (**b**) and *z* axis (**c**). The networks on the left represent graphene, and the balls on the right represent adjacent Al atoms, which are colored by per-atom stress (unit: bar × Angstrom^3^).

**Figure 7 materials-15-02644-f007:**
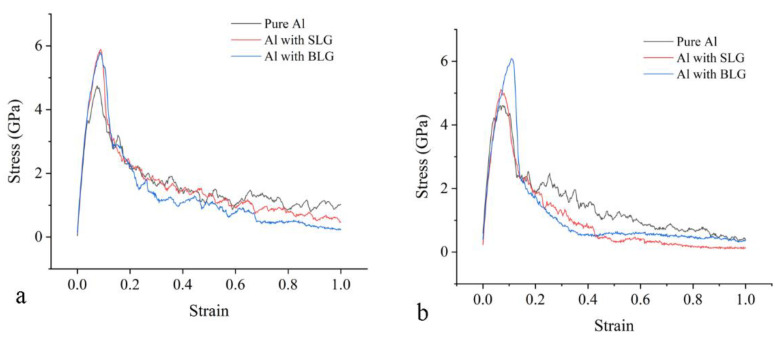
The stress–strain curves in tensile simulations of the sintered models. (**a**) The models are stretched in the *y* direction; (**b**) the models are stretched in the *z* direction.

**Table 1 materials-15-02644-t001:** Experimental hardness of Al–graphene composites sintered by powder metallurgy.

Graphene Type	Al Matrix	Graphene Content	The Improvement in Hardness	Research Group
Graphene nanoflakes	Al6061	0.4 wt%	8.3%	Kumar et al. [7]
Reduced graphene oxide	Al	0.3 wt%	32%	Liu et al. [8]
Graphene nanoflakes	Al	0.15 wt%	43%	Liu et al. [8]
Reduced graphene oxide	Al	0.3 wt%	17%	Li et al. [6]
Graphene oxide	Al7075	1 wt%	15%	Shin et al. [9]
Graphene nanoflakes	Al	0.3 wt%	11.8%	Rashad et al. [11]

**Table 2 materials-15-02644-t002:** Analysis on sintered structures.

Model	Final Volume (nm^3^)		The Number of Atoms Analyzed by CNA	
	Entire System	Al Atoms	f.c.c. Atoms	h.c.p. Atoms
Pure Al	759.6	759.6	38,944	4606
Al with SLG	753.4	744.8	35,216	3752
Al with BLG	759.0	739.0	31,687	6454

## Data Availability

Not applicable.

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
