# Peer review of "Atomistic Study on the Sintering Process and the Strengthening Mechanism of Al-Graphene System"

_materials, 2022, doi:10.3390/ma15072644_

Round 1

Reviewer 1 Report

-Provide more quantitative results of your study in the abstract
-Line 51-53 - the literature review is fine with me. The statement as well, maybe, just as a suggestion provide a graph that shows the graphene content of the different studies done before and the increased strength in percent of the material. I think this would be a wonderful addition to illustrate your point
-In the description of your experiment you can again draw a little graph that shows temperature, pressure, time
-the density could further be increased through pressure variations - there is an interesting recent study where this technology is proposed for diffusion bonding - that is essentially sintering of plates - Ref: https://doi.org/10.3390/met11020323
-really make sure that it is clear what was simulated and what is actually experimental work - this should also be indicated in caption of figures such as Fig. 7
-in the conclusion as well provide more quantitative results of your study, at present it is not really clear to me what improvement this simulation exercise really brought. Maybe try to also verify your simulation against actual experimental data - from you or another research group.

Reviewer 2 Report

In the presented manuscript entitled as "Atomistic study on the sintering process and the strengthening mechanism of Al-Graphene system" authors perform large-scale MD simulations to investigate the role of graphene and bigraphene layers on the formation of Al nanoparticles during sintering.

Such topic is highly important in the view of development of the filed of new strength and light materials of new generation. Graphene is well-known material which is together with h-BN the main candidate for application in this field. 

On the the important result of this study is that graphene lead to the formation of the smaller Al nanoparticles during sintering and prevent agglomeration with following formation of rather large particles. 

Unfortunately, authors missed a little in the introduction the previous theoretical studied of using graphene for increasing of the strength of Al and light metal materials. Some of these:

https://doi.org/10.1039/C6NR07206B

During the manuscript reading a few things remained unclear to me:

1) In the abstract the sentence: "Because these models are carefully designed to avoid the deformation of graphene caused by direct force, the strengthening mechanism close to experimental theory can be studied dynamically." What is the meaning of it? Does it mean that the simulations model was fit in a specific way to avoid graphene deformation, but in reality graphene layer could be significantly deformed? Does it mean that during calculations you did not take into account graphene deformations? Please clarify this.

2) The last paragraph of introduction starting from the reference 12 up to 17 is presented like sequential listing of what has been done, without any details and the main conclusion. Looks very foreign in this text.

3) In the "Model methods" section you wrote about the fitting of Morse potential for description of Al-C interaction. Can you present any comparison data with DFT calculations or experiment data to validate this statement?

4) Figure 7: What happened with the atomic structure in the range of strain from 0.2? Detail description will be very useful for understanding of the process.

According above mentioned I could state that presented manuscript could be published in MDPI Materials after minor corrections.

Round 2

Reviewer 1 Report

Excellent - all comments have been addressed - I will recommend this work for publication.